# Sustainable Development in World Trade Law: Application of the Precautionary Principle in Korea-Radionuclides

**Yan Cai and Eunmi Kim \***

Department of International Trade, College of Commerce, Chonbuk National University, Jeonju 54896, Korea; rebecca121@naver.com

\* Correspondence: eunmi@jbnu.ac.kr; Tel.: +82-632703016

**Abstract:** Sustainable development (SD) is one of the objectives of the World Trade Organization (WTO), and in order to achieve SD, the precautionary principle (PP) is one of the most appropriate means that can be used. This study aims to explore whether the WTO promotes SD through its legal interpretation of the PP and to provide suggestions for realizing the balance between trade liberalization and sustainable development in the WTO. To this end, this study conducts a case analysis on the *Korea-Import Bans, and Testing and Certification Requirements for Radionuclides* (DS495) dispute from legal and political-economic perspectives, and finds that the WTO's rigorous examinations of the four requirements presented in Article 5.7 of the Agreement on the Application of Sanitary and Phytosanitary Measures (the SPS Agreement) remain a significant impediment for the incorporation of the PP into WTO jurisprudence, and can also cause systemic problems. This study suggests that efforts from three dimensions—the WTO adjudicating parties, the government, and the lobbying groups of importing countries—are required to promote SD in the WTO.

**Keywords:** sustainable development; precautionary principle; world trade law; SPS Agreement; Korea-Radionuclides

---

## 1. Introduction

The preamble of the Marrakesh Agreement Establishing the World Trade Organization highlights sustainable development (SD) as one of its far-reaching objectives running through all multilateral trade agreements [1]; in order to achieve SD, the precautionary principle (PP), which is regarded as a "prerequisite" or "key foundation" for SD, is one of the most appropriate means that can be used [2–4].

The PP constitutes a core element of numerous national, regional, and international environmental laws involving scientific uncertainty and irreversible risk [5]; however, there is no universally accepted definition of the PP, nor is there a consensus regarding its legal status as a general principle of law. Some researchers have suggested that the PP has ripened into an enforceable customary international law which no country can disavow, particularly due to its application in numerous multilateral treaties [6–9]. By contrast, other researchers have stated that the PP is neither recognized as a general principle in international law nor a binding policy guideline, due to the legal weakness of the PP in world trade law [10–13]. In the WTO's multilateral trade agreements, there are no explicit mentions of the word "precaution", aside from an indirect equivalence in Article 5.7 of the WTO Agreement on the Application of Sanitary and Phytosanitary Measures (the SPS Agreement), which allows for the adoption of provisional sanitary and phytosanitary (SPS) measures in the case of insufficient scientific evidence.

This study aims to explore whether the WTO promotes SD through its legal interpretations of the PP and to provide recommendations for realizing the balance between trade liberalization

and sustainable development in the WTO. To this end, this article proceeds as follows: Chapter 2 illustrates the incorporation of the SD and PP in environmental laws as well as their rationales in world trade law. Sustainable development is defined as "development which meets the needs of the present without compromising the ability of future generations to meet their own needs" [14]. There are numerous principles to ensure SD, such as principles of "user pays", "polluter pays", and "subsidiary principle". The PP is one of them [15]. It can be "a valuable aid" to SD [16]. Chapter 3 briefly reviews all precedents related to the consistency of Article 5.7 of the SPS Agreement. Among them, *Korea-Import Bans, and Testing and Certification Requirements for Radionuclides (Korea-Radionuclides)*, whose Panel Report was circulated very recently on 22 February 2018, was analyzed from legal and political-economic perspectives. Chapter 4 identifies systemic problems of the disparity between panels and AB in interpreting PP, and suggests that efforts from three dimensions—the WTO adjudicating parties, the government, and the lobbying groups of importing countries—are required to promote SD in the WTO.

## 2. Rationales of Sustainable Development in WTO

International trade is a powerful ally of SD [17]. Back in 1992, Agenda 21, adopted in the United Nations Conference on Environment and Development recognized that a multilateral trading system could contribute to SD [18]. At that time, the multilateral trading system came under the General Agreement on Tariffs and Trade (GATT), the WTO's predecessor. When the WTO was established in 1995, WTO Members included direct references to the objective of SD [17]. The preamble of the Marrakesh Agreement Establishing the World Trade Organization (the WTO Agreement) mandated that it should be " . . . allowing for the optimal use of the world's resources in accordance with the objective of sustainable development, seeking both to protect and preserve the environment and to enhance the means for doing so . . . " [19]. This mandate was reaffirmed in the 2001 Doha Ministerial Declaration which declared that trade liberalization should contribute to SD. Practically, WTO Members set up SD as a central principle for new negotiations throughout all Doha negotiations [20]. Although the stated importance indicates that SD wields great influence, due to the non-binding nature of preambular language and the comatose state of the Doha Round, SD does not form a legal rule. It is only a general principle of the WTO's legal framework [21]. This is not to say SD has no formal influence [21]. Instead, it not only serves as guidance for the implementation of WTO agreements, such as the SPS Agreement, but also serves as a justification for decisions by dispute settlement bodies [1].

There are a number of mechanisms that can be used to promote SD, one of which is the PP; generally, the PP is defined as the concept in which, when an activity causes serious or irreversible threat of harm to the environment or public health [22], measures can be taken even in situations of scientific uncertainty [23]. Since the PP can promote the realization of SD by limiting the irreversible risks, it received broad recognition in many national, regional, and international environmental laws.

At the national level, Germany was the first country to adopt the concept of PP by legislating the German Federal Immission Control Act in 1974 with the purpose of taking precautions against the emergence of any potential harmful effects on the environment [24]. Germany went on to introduce the PP to the whole European Community during several conferences held in Bremen (1984), London (1987), Hague (1990), and Esbjerg (1995) regarding the protection of the North Sea [25]. The European Union (EU) continued this trend in proactively regulating uncertainty; finally, the EU member countries reached a consensus on the adoption of the PP in 1992 as an overarching environmental policy in Article 174.2 of the Treaty Establishing the European Community: "Community policy on the environment . . . shall be based on the precautionary principle . . . " [26]. Along with the EU, the United States (US) also began to include precautionary elements in its environmental regulatory regime, such as the US Food Quality Protection Act and the Sustainable Fisheries Act [27]. Further, Korea adopted the PP in Article 8 of the Framework Act on Environmental Policy which states "The State and local governments shall exert preferential efforts for a precautionary pollution control . . . " [28]. More examples of the

adoption of PP can be found in the domestic laws of Australia, Canada, New Zealand, Chile, India, and Pakistan, among others [16].

The widespread adoption [29–31] of the PP in these domestic and regional laws has contributed to its incorporation into numerous international instruments of both soft and hard law [9]. The most prominent statement of the PP in a soft law instrument [22] can be found in Principle 15 of the Rio Declaration on Environment and Development of 1992: "In order to protect the environment, the precautionary approach shall be widely applied by States according to their capabilities. Where there are threats of serious or irreversible damage, lack of full scientific certainty shall not be used as a reason for postponing cost-effective measures to prevent environmental degradation" [32]. Since then, the application of PP has proliferated a vast number of hard law instruments such as the United Nations Framework Convention on Climate Change, Convention for the Protection of the Marine Environment of the North-East Atlantic, the Cartagena Protocol on Biosafety to the Convention on Biological Diversity, and the Stockholm Convention on Persistent Organic Pollutants [9]. For example, the Cartagena Protocol on Biosafety as a supplementary agreement to the Convention on Biological Diversity provides for PP to be applied to the importation of living modified organism in Article 11.8: "Lack of scientific certainty due to insufficient relevant scientific information and knowledge ... shall not prevent that Party from taking a decision, as appropriate, with regard to the import of that living modified organism ... " [33]. By requiring that activities should not be allowed if there is no guarantee of no harm, this provision prescribes a strong PP [34]. In addition to the Cartagena Protocol on Biosafety, this principle has also been reflected in the WTO's multilateral trade agreements, particularly, in the SPS Agreement. However, as they have been designed to serve different purposes, the SPS Agreement prescribes a weak PP [34].

The SPS Agreement provides for an entry point for PP in Article 5.7, albeit with qualifications [22]. Article 5.7 of the SPS Agreement allows for the adoption (in the first sentence) and maintenance (in the second sentence) of provisional SPS measures provided that the following four requirements are fulfilled: (i) a Member may adopt provisional measures in such cases where "relevant scientific evidence is insufficient" to conduct a risk assessment (R1); (ii) the provisional measures must be adopted on the basis of "available pertinent information" (R2); (iii) the Member maintaining the provisional measures shall "seek to obtain the additional information necessary for a more objective assessment of risk" (R3); and (iv) the Member maintaining the provisional measures shall review that measure "within a reasonable period of time" (R4). This provisional approach to avoid risks with insufficient scientific evidence is similar to the "better safe than sorry" wisdom of the PP, which underpins uncertainty and irreversibility [13].

Not only the SPS Agreement, but also Article XX of GATT, appears to take account of elements of the PP [22]. For example, in *EC–Asbestos*, the Appellate Body (AB) opened an entry point for elements of the PP under Article XX (b) of GATT and allowed a high degree of deference to domestic decision-making on appropriate level of risk. It found that WTO Members had the right to determine the level of protection of health that they considered appropriate [35]. However, a further analysis on the application of PP in the GATT is not within the scope of this study. Thus, the following sections will only focus on the application of PP in Article 5.7 of the SPS Agreement.

## 3. Case Analysis

### 3.1. WTO Jurisprudence on the PP

The WTO adjudicating bodies have examined the PP on a case-by-case basis [9] and WTO jurisprudence on it is as follows. As shown in Table 1, disputes over the consistency of the provisional measure in the WTO have existed for more than 20 years, among which most cases did not mention PP as it is not in the wording of Article 5.7 of the SPS Agreement.

**Table 1.** WTO dispute cases related to Article 5.7 of the Sanitary and Phytosanitary (SPS) Agreement.

| Year | | Dispute Number | Complainant | Short Title | Is the SPS Measure Consistent? | |
| --- | --- | --- | --- | --- | --- | --- |
| Panel Report | AB Report | | | | Panel Finding | AB Finding |
| 1997 | 1998 | DS26 DS48 | United States Canada | EC-Hormones | Inconsistent | No appeal |
| 1998 | 1999 | DS76 | United States | Japan-Agricultural Products II | R3 [N], R4 [N] | R3 [N], R4 [N] |
| 2003 | 2003 | DS245 | United States | Japan-Apples | R1 [N] | R1 [N] |
| 2006 | No appeal | DS291 DS292 DS293 | United States Canada Argentina | EC-Approval and Marketing of Biotech Products | R1 [N] | No appeal |
| 2008 | 2008 | DS320 DS321 | European Communities European Communities | US-Continued Suspension Canada-Continued Suspension | R1 [N] | No finding |
| 2015 | No appeal | DS447 | Argentina | US-Animals | R3 [N], R4 [N] | No appeal |
| 2016 | 2017 | DS475 | European Union | Russia-Pigs | R1 [N], R2 [N], R3 [N], R4 [N] | No appeal |
| 2018 | Not completed | DS495 | Japan | Korea-Radionuclides | R1 [N], R2 [N], R3 [Y], R4 [N] | Not completed yet |

Notes: The four requirements of Article 5.7 are marked as R1, R2, R3, and R4. [Y] = consistent with the requirement, [N] = inconsistent with the requirement. Source: Compiled by authors from information on the WTO website [36].

One exception is the *EC- Hormones* case, which is the only one that explicitly discussed the "precautionary principle" as one of the core issues [13,37]. In this dispute, the EC invoked the PP to justify that its SPS measures were based on a risk assessment; nevertheless, the Panel found that the EC's import ban was not based on a risk assessment, since the EC had explicitly stated that its SPS measures were not provisional [38] (paras. 8.157–8.158). In the appeal proceeding, the EC did not appeal Article 5.7, but instead argued that the PP had become "a general customary rule of international law" or at least "a general principle of law"; in response, the US and Canada took the view that the PP was more an "approach" than a "principle" and had not yet been incorporated into the corpus of public international law [39] (paras. 120–121). The Appellate Body (the AB) considered that it could be imprudent to make a definitive finding regarding the legal status of PP because it continues to be a subject of debate; however, it did acknowledge that the PP could find its reflection in Article 5.7 of the SPS Agreement [39] (paras. 121–124).

Since the AB in *EC- Hormones* acknowledged the relation between PP and Article 5.7, WTO Members have been more inclined to invoke Article 5.7 than have direct recourse to the PP in justifying their provisional SPS measures. For instance, in *Japan-Agricultural Products II*, Japan invoked Article 5.7 and argued that its import prohibition for US agricultural products met the requirements therein; in response, the US argued that the available scientific evidence was not insufficient to perform a risk assessment and Japan did not review its SPS measures within a reasonable period of time. The Panel, ruling in favor of the US, found that Japan had not met the requirements contained in the second sentence of Article 5.7, since Japan did not seek to obtain the necessary additional information and review its SPS measures [40] (paras. 8.50–8.59). The AB upheld the Panel's finding and emphasized that the four requirements set out in Article 5.7 were equally important and cumulative in nature; that is, the provisional measure would be inconsistent with Article 5.7 whenever one of these four requirements was not fulfilled. In this case, since the AB found that Japan did not meet R3 and R4, it was unnecessary to address R1 and R2 [41] (paras. 89–94). This kind of judicial economy was also exercised in *Japan-Apples*, *EC-Approval and Marketing of Biotech Products*, *US/Canada-Continued Suspension* and *US-Animals*, where the panels and AB did not address the other requirements once they

found that one or two of the four requirements were not met; however, this practice did not continue in *Russia-Pigs* and *Korea-Radionuclides*, where all of the four requirements were examined by the panels.

It is worth noting that the interpretation of PP is not uniform between panels and the AB. This disparity has been shown in the adjudications of SPS disputes [42]. Early in *EC-Hormones*, the AB had made it clear that "responsible, representative governments commonly act from perspectives of prudence and precaution where risks of irreversible, e.g., life-terminating, damage to human health are concerned" [39] (para. 124). This statement indicates that the AB appears to allow WTO Members the policy space necessary to operationalize the PP [22]. This approach of interpretation has become more evident in *US/Canada-Continued Suspension* [42]. According to the Panel's interpretation, the condition to "make relevant, previously sufficient, evidence now insufficient" was that "there must be a critical mass of new evidence and/or information that calls into question the fundamental precepts of previous knowledge and evidence" [43] (para. 7.648). However, the AB reversed the Panel's finding and criticized that such a requirement was too inflexible since it led to a "paradigm shift" [44] (para. 703). Furthermore, the AB pointed out that Members should be allowed to "take a provisional measure where new evidence from a qualified and respected source puts into question the relationship between the pre-existing body of scientific evidence and the conclusions regarding the risks" [44] (para. 703).

Although the AB deserves credit for moving further away from a strict interpretation of Article 5.7 [22], it did not make a final finding as to the consistency or inconsistency of the EC's provisional measures with Article 5.7 [44] (para. 736), leaving this fundamental question open [45]. Here, we conduct a case analysis on *Korea-Radionuclides* for the following reasons: First, it is the latest case related to Article 5.7, so a study on it can contribute to a useful understanding of the current state of WTO ruling on Article 5.7; second, it is the first WTO dispute in which the radioactive issue was discussed, so a study on it can serve as guidance for other Members if another nuclear leakage accident happens in the future.

*3.2. Factual Aspects of Korea-Radionuclides*

On 11 March 2011, a huge amount of radioactive materials was released into the atmosphere, land, and ocean from the Fukushima Dai-ichi Nuclear Power Plant (FDNPP), operated by the Tokyo Electric Power Company, as a result of a reactor accident following the Great East Japan Earthquake and a subsequent devastating tsunami [46] (paras. 2.40–2.43). The Korean government responded to the FDNPP accident by imposing a variety of import control measures on certain Japanese fishery products. On 21 May 2015, Japan requested consultations with Korea, but the two countries failed to solve their disputes; then, on 20 August 2015, Japan requested the establishment of a panel and challenged three of Korea's measures, as shown in Table 2.

**Table 2.** Korea's measures challenged by Japan.

| Type | Date of Imposition | Content of the Measure | Products Covered | Applied Prefectures |
|------|--------------------|------------------------|------------------|---------------------|
| T1 | 1 May 2011 | Additional radionuclides must be tested for when trace amounts of caesium or iodine are detected | All agro-forestry products, processed foods, food additives, and health functional foods | All 47 prefectures |
| T2 | 2 May 2012<br>22 June 2012<br>27 August 2012<br>9 November 2012 | Product-specific ban<br>Product-specific ban<br>Product-specific ban<br>Product-specific ban | Pacific cod<br>Pacific cod, Alaska pollock<br>Pacific cod<br>Pacific cod | Miyagi, Iwate<br>Fukushima<br>Aomori<br>Ibaraki |

**Table 2.** *Cont.*

| Type | Date of Imposition | Content of the Measure | Products Covered | Applied Prefectures |
|------|--------------------|------------------------|------------------|---------------------|
| T3 | 9 September 2013 | Blanket import ban | 28 fishery products | Aomori, Chiba, Fukushima, Gunma, Ibaraki, Iwate, Miyagi and Tochigi |
| | 9 September 2013 | Additional radionuclides must be tested for when more than trace amounts of caesium or iodine are detected | All fishery and livestock products | All 47 prefectures |

Source: Adapted from the Panel Report of the Korea-Radionuclides [46] (para. 2.115).

The Panel Report, which found Korea's import restriction measures to be in violation of WTO rules, was circulated on 22 February 2018; however, Korea decided to keep the ban in place and notified the dispute settlement body (DSB) of its decision to appeal in April 2018. The AB was supposed to have circulated its report no later than 60 days after the appellants' appealing decisions, but failed to do so due to the current vacancies of AB members and enhanced workload, and the circulation date of the AB Report remains pending [47].

There are five main controversial issues in the Panel Report, the first of which is the burden of proof. According to Korea, the burden of demonstrating compliance with Article 5.7 was on Japan, and Korea argued that the Panel must presume that Korea's measures fall within the scope of Article 5.7 because Japan did not raise Article 5.7 in its claims. However, Japan contended that, as it was Korea that invoked Article 5.7, Korea thus bore the burden of proving whether the four requirements of Article 5.7 had been satisfied. The Panel ruled in favor of Japan and found that Korea had to bear the burden of proof [46] (paras. 7.70–7.75); next, it examined whether Korea proved itself to have satisfied all of the four requirements set forth in Article 5.7.

Regarding R1, Korea argued that the information regarding the extent of the radionuclides released during and after the FDNPP accident was insufficient for conducting a proper risk assessment (RA) [46] (para. 7.79). In response, Japan cited a number of reports published by international organizations and argued that Korea did not consider the available scientific evidence and even seemed intent on ignoring it [46] (para. 7.82). The Panel agreed with Korea that T1 satisfied R1, because Korea was uncertain about the extent of hazards immediately following the FDNPP accident and mirrored Japan's emergent measures at that time; however, the Panel found that T2 and T3 failed to fulfill R1 because some estimates regarding the leakage amounts were publicly available [46] (para. 7.91).

Regarding R2, Korea referred to (i) public concern to the discharged contaminated water, (ii) uncertainties of the evolvement of the situation in Japan, (iii) inability to predict the future development based on Japan's data, and (iv) the Codex Standard as "available pertinent information" and argued that its SPS measures were based on them [46] (paras. 7.97 and 7.99). In response, Japan argued that the mere list of information did not prove that Korea's measures were based on that information [46] (para. 7.97). The Panel found that T1 and T2 satisfied R2, since Korea referred to the guideline levels of the Codex Standard for radionuclides [46] (para. 7.98); next, the Panel recalled the AB's interpretation of the term "based on" in *EC-Hormones* and confirmed that "a thing is commonly said to be 'based on' another thing when the former 'stands' or is 'founded' or 'built' upon or 'is supported by' the latter" [39] (para. 163). The Panel considered that Article 5.7 focused on basing the measure on science, so (i), (ii), and (iii) were not the type of available information that was pertinent. As for (iv), the Panel noted that the Codex Standard merely established tolerance levels below which food can be safely traded, and did not call for the adoption of import bans, so Korea's blanket import ban in 2013 could not be based on the Codex Standard; accordingly, the Panel found that T3 did not satisfy R2.

Regarding R3, Japan argued that Korea had not sought to proactively obtain additional information since its adoption of the measures at issue; nevertheless, Korea submitted a variety of recorded evidence to prove that it indeed did seek to obtain new information through numerous communications with Japan's authorities. Based on these shreds of evidence, the Panel accepted Korea's arguments and found that Korea fulfilled its obligation to seek additional information [46] (paras. 7.103 and 7.107).

Regarding R4, Japan argued that the Korean government merely planned a schedule for reviewing its measures in February 2014 but failed to implement this schedule as planned. In response, Korea claimed that it did undertake all of the review steps except for completing a report of risk assessment. The Panel found that Korea failed to satisfy R4 since it did not review its measures within a reasonable period of time [46] (paras. 7.104–7.107).

In conclusion, the Panel found that, since Korea's SPS measures did not fulfill all of the four cumulative requirements, its measures did not fall within the scope of Article 5.7. Table 3 summarizes the Panel's findings.

**Table 3.** Summary of the Panel's findings.

| Requirement / Type of Measures | T1 | T2 | T3 |
|:---:|:---:|:---:|:---:|
| R1 | O | X | X |
| R2 | O | O | X |
| R3 | | O | |
| R4 | | X | |
| Cumulative assessment | | X | |

Notes: O = fulfillment, X = non-fulfillment. Source: Compiled by authors based on the Panel Report of the Korea-Radionuclides [46].

### 3.3. Legal Observations

The *Korea-Radionuclides* dispute raises three important issues about the interpretation of Article 5.7, that is, burden of proof, insufficient scientific evidence, and review of the measure. First, the *Korea-Radionuclides* dispute revealed a clear tension inherent in the SPS Agreement regarding the assignment of the burden of proof. The Panel of this dispute based its reasoning on the panel decision in *EC-Approval and Marketing of Biotech Products*, which established that it was incumbent on the complaining party to demonstrate that the challenged SPS measures were inconsistent with at least one of the four requirements set forth in Article 5.7. However, once the complaining party established a *prima facie* case of inconsistency with Article 5.7, the burden of proof would shift to the defending party, which would then have to prove that the available scientific evidence was insufficient [48] (para. 7.2979). The problem is that it is still ambiguous whether and when a *prima facie* case has been made; generally, this relies on the exclusive competence of a panel which does not have to make an explicit statement involving this [49].In *Korea-Radionuclides*, the Panel ruled that Japan established a *prima facie* case simply because Japan argued that the party who invoked Article 5.7 should bear the burden of proof, indicating that the Panel of this dispute allowed a rather weak *prima facie* case in shifting the full burden of proof to Korea.

Second, Korea, by referring to numerous reports and articles which suspected that more radioactively contaminated water was leaking than disclosed, argued that the scientific evidence regarding the extent of existing contamination was insufficient to conduct an RA [46] (para. 7.90). The Panel agreed with Korea in this respect, but mentioned that "scientific evidence need not be 100% complete or perfect to be sufficient to form the basis for an objective assessment of the risk" [46] (para. 7.89). In fact, the Panel's assessment dismisses the logic indicating that it is the existence of sufficient scientific evidence that constitutes the premise to conduct a proper RA; if the scientific

evidence is allowed to be imperfect, it implies that imperfect scientific evidence can also constitute the basis for a proper RA. However, the Panel did not clearly explain the extent of allowable imperfection; as a result, the ambiguity may create a legal advantage for the party who tries to persuade the Panel that an RA can be conducted, while it may impose a heavy burden on the party who tries to claim the contrary.

In terms of the relevance of uncertainty regarding the amounts and share of different radionuclides to an RA of food products from Japan, the Panel turned to experts for technical advice. The experts declared that the best way to know the actual levels of radionuclides in foods was by performing measurements on them instead of on the environment, and that this kind of food-test could be conducted using existing technology. Therefore, the experts who were invited by the Panel unanimously agreed that uncertainties about the total amounts of continued release to the environment could not prevent a sound RA to levels of contamination in foods [46] (paras. 7.92–7.93). Moreover, the Panel recalled that Korea's SPS measures were designed to protect Korean consumers from food-exposure rather than environment-exposure to radionuclides. Since Korea established its own tolerance levels for caesium and applied the Codex guideline levels for other radionuclides, the Panel assumed that the risk of food-exposure could be assessed [46] (para. 7.93).

Before analyzing rulings of the Panel, it is worth explaining the legal context in which Article 5.7 operates. First, Article 3.3 of the SPS Agreement confers upon Members the right to determine their own level of protection, including "a higher level of sanitary or phytosanitary protection than . . . international standards, guidelines or recommendations", under the condition that "there is a scientific justification, or as a consequence of the level of sanitary or phytosanitary protection a Member determines to be appropriate in accordance with the relevant provisions of paragraphs 1 through 8 of Article 5". Second, while Article 2.1 of the SPS Agreement allows Members to take SPS measure, Article 2.2 mandates that this can be done with a considerable amount of scrutiny [42] by requiring Members to ensure that SPS measure "is applied only to the extent necessary . . . and is not maintained without sufficient scientific evidence, except as provided for in paragraph 7 of Article 5". Thus, Article 5.7, which allows Members to adopt provisional SPS measures in cases where "relevant scientific evidence is insufficient", operates as a "qualified exemption" from the obligation under Article 2.2 [41] (para. 80).

The assessments of the Panel and experts indicate that, even in the situation where the extent of release of radionuclides to the environment is uncertain, Korea has to import Japan's fishery products first, and then if Korea wants to take provisional SPS measures, it must conduct a food-test to prove that the available scientific evidence is insufficient to conduct an RA on the fishery products. The Panel's approach to interpretation appears to be contradictory because it is difficult to provide legal evidence of the insufficiency of scientific evidence, and this kind of paradox has also been criticized by a number of scholars, such as Vecchione [45], Perez [50], and Scott [51]. The evidence required is burdensome in scientific terms because it corresponds to a "paradigm shift", that is, the disruption of already available scientific evidence contained in the existing international standards. It is very difficult for Korea to satisfy R1 because in the logic of the SPS Agreement, it is a deviation from an international standard. Although the AB has repeated its long-standing position that Members have the right to set their appropriate level of protection [44] (para. 692) and, according to Wagner [22,42], the AB has allowed WTO Members to "rely on minority scientific opinions when determining whether there was insufficient scientific evidence in order to justify more stringent measures", the Panel's ruling highlights the continued reluctance of the panels to give governments considerable discretion to justify their provisional measures when there is insufficient evidence to conduct a risk assessment [42].

Aside from the existing contamination, Korea also referred to the insufficiency of scientific evidence related to potential future contamination. The Panel agreed with Korea that it was unknown whether an additional accident could occur in the future; however, Article 5.7 was not meant to address this kind of permanent uncertainty [46] (para. 7.95). The Panel's interpretation suggests a

narrow understanding of the concept of insufficient scientific evidence. However, Korea's provisional measures can be long-running as long as new evidence is being sought.

Third, we noted that the Korean government did not provide a final report of its reviewing results and thus failed to satisfy its obligation to review the measure within a reasonable period of time. In the text of the SPS Agreement, there exists no explicit definition of what constitutes "a reasonable period of time", so the Panel referred to the interpretations of precedents. In *Japan -Agricultural Products II*, the AB considered that this had to be "established on a case-by-case basis and depends on the specific circumstances of each case" [41] (para. 93); in *EC-Approval and Marketing of Biotech Products,* the Panel interpreted the term "reasonable period of time" as "undue delay" [48] (paras. 7.1495–7.1497); in *US-Animals*, the Panel noted that a "reasonable period of time" meant "as quickly as legally possible while accepting legitimate reasons for delay" [52] (para. 7.301). This indicates that precautionary measures cannot be adopted for an unlimited period of time without being subjected to review; however, the length of time that is considered to be too long is within the discretion of the panels.

Regarding why a final RA report was not finished, Korea did not provide any legitimate explanations; however, Japan argued that a Korean/Civilian Expert Group conducted an RA but the findings would not support Korea government's measures, so Korea rejected to disclose it [46] (para. 7.103). For its part, Korea argued that this Civilian Expert Group did not represent the Korean government, and suspended its activities in June 2015 after the onset of consultations in the WTO [46] (para. 7.105). From the perspective of the Panel, even though the activities of the Korean/Civilian Expert Group somehow constituted a review of Korea's SPS measures, the Panel could not reach a conclusion that its actives represented an official review of Korean government in light of Korea's clarification of its role.

Furthermore, Korea did not provide any justified reason for the suspension of the activities of the Korean/Civilian Expert Group; in this regard, the Panel noted that the mere onset of consultations in May 2015 did not justify the incompliance with the reviewing obligation [46] (paras. 7.106–7.107). Therefore, Korea's explanation that Japan's complaint constitutes the reason for the suspension of its review activities was not persuasive. In order to ensure the legitimacy of the Korean government's provisional SPS measures based on Article 5.7, it should continue to carry out reviewing activities, even after the establishment of the Panel.

*3.4. Political-Economic Observations*

3.4.1. Japan's WTO Complaint against Korea

Since the FDNPP accident, 54 countries and regions have adopted restrictive or prohibitive measures on imported Japanese food products in order to protect public health; among them, 20 countries including Korea, China, and the US, have imposed import bans [53]. However, Korea is the only country that Japan has taken to the WTO. The following three reasons may contribute to Japan's complaint: Possible economic benefits, domestic lobbying pressure, and the political warning effect to other countries.

First, conventional wisdom holds that countries are inclined to initiate "high stakes" disputes, that is, countries tend to challenge trade barriers in cases where the potential economic gains are sizeable [54–57]. As shown in Figure 1, there has been a general upward trend in Japan's exports over the last 17 years, with small fluctuations around 2008 (global financial crisis) and 2011 (FDNPP accident). Prior to the 2011 FDNPP accident, Korea was the number one export destination for Japan's fishery products, accounting for 20% of total exports; however, this share fell to 8% following the FDNPP accident.

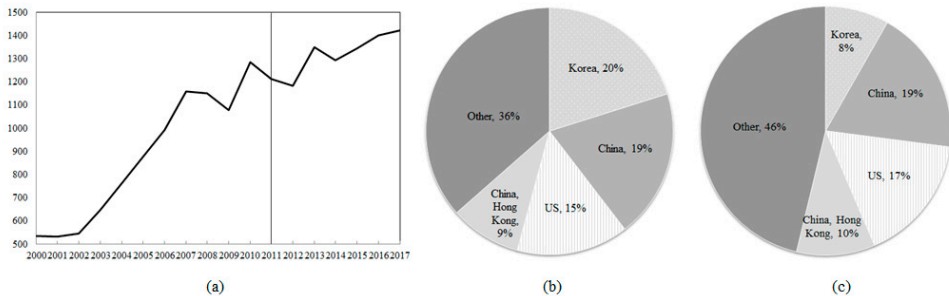

**Figure 1.** Japan's exports of fishery products to the world by value (millions of dollars). (**a**) Trend from 2000 to 2017. (**b**) Share prior to the accident. (**c**) Share after the accident. Source: Authors' calculations based on the UN Comtrade Database [58].

After the accident, despite the fact that Japan's other top export destinations have also taken import-restrictive measures regarding its fishery products, Korea has been the only country which has presented a remarkable decreasing trend of imports from Japan. As shown in Figure 2a, Korea's annual import value from Japan decreased by 38% after the 2011 accident; by contrast, Japan's other top export destinations, such as China and the US, increased their imports by 35% and 69%, respectively. However, the downward trend of Korea's imports from Japan had already begun as early as 2008; it did not suddenly begin to decline after the 2011 accident. As shown in Figure 2b, Japan's export of fishery products to Korea began to decrease in 2008, likely due to the global financial crisis and extended demand from China; then, the downward trend continued after the FDNPP accident in 2011 and reached its lowest point before Japan lodged a complaint with the WTO in 2015. This suggests that Korea's imposition of import-restrictive measures after the 2011 accident was not the only reason for its decreased import of fishery products from Japan, and as such, further investigation is needed.

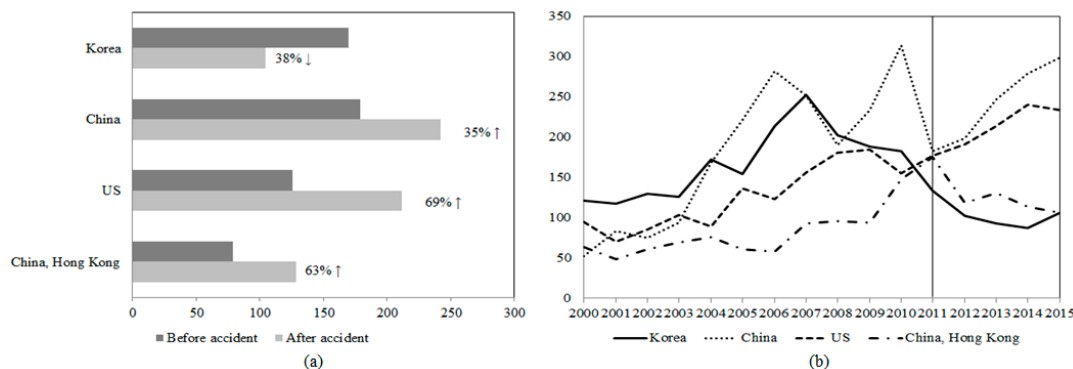

**Figure 2.** Japan's export of fishery products to top export destinations by value, 2000–2015 (millions of dollars). (**a**) Comparison of change. (**b**) Trend of change. Source: Authors' calculations based on the UN Comtrade Database [58].

Another explanation for Korea's decreased importing from Japan may lie in its increased domestic supply. As shown in Figure 3a, Korea's average annual domestic consumption and production have increased by 739,273 tons and 751,084 tons, respectively, indicating that Korea's decreased importing was mainly compensated for by its increasing domestic production. Figure 3b shows that there is a trend for domestic production accounting for more domestic consumption; therefore, Japan's win over Korea at the WTO may not necessarily promote its exports to Korea nor promise tangible economic benefits, and in fact, the confrontation between the two countries in WTO will make their diplomatic relations more tense.

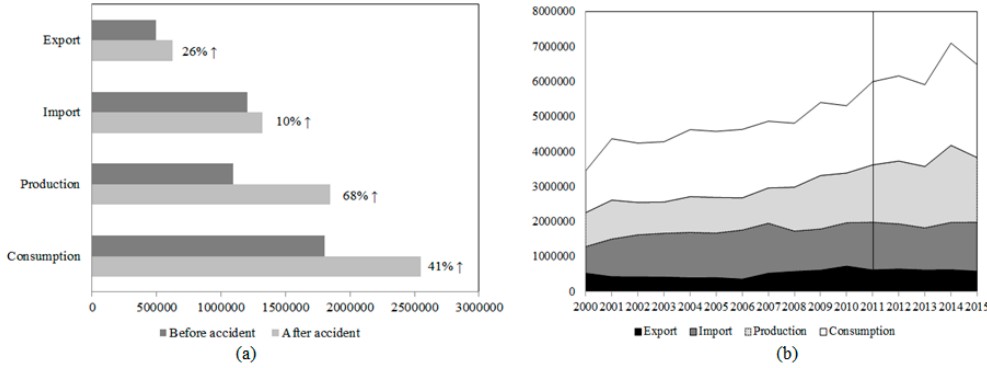

**Figure 3.** Korea's market for fishery products by amount (unit: ton): (**a**) Comparison of change; (**b**) trend of change. Source: Authors' calculations based on the Fishery Commodities Global Production and Trade database [59].

Second, previous empirical evidence suggests that domestic industry pressure can also push government leaders to launch a trade complaint at the WTO [60,61]. Given the fact that Korea used to be the top export destination for Japan's fishery products before the 2011 accident, it is too attractive a market for the Japanese fishery industry to give up on. Therefore, it is highly possible that actors within Japan's fishery industry lobbied the government to file a complaint against Korea, despite the fact that lobbying details are not publicly available. However, there is little possibility for the domestic industry to gain any expected benefits because it is proven above that the main reason for Korea's decline in importing from Japan is that Korea's increasing domestic supply means that it can now meet its domestic consumption. Accordingly, rather than challenging Korea's SPS measures at the WTO, it would be wiser for Japan to design an appropriate export strategy by exploring the reasons behind Korea's continued increasing trend in domestic supply, such as possible government subsidies, the operational status of Korea's fishery industry, etc.

Third, Japan's win in the WTO dispute can act as a warning to other countries who still maintain their import-restrictive measures to Japan's products. As aforementioned, no country has ever successfully applied Article 5.7 of the SPS Agreement in justifying its provisional SPS measures, implying that the anticipated likelihood of Japan's win is high. However, this cannot fully explain why Japan only complained about Korea and not about its other top export destinations. For instance, China was the second largest export destination for Japan's fishery products before and after the 2011 accident, but it did not become the target of a complaint from Japan. One reason for this may be that Japan's exports of fishery products to China increased after the FDNPP accident (as shown in Figure 2), so it had no economic conflicts of interest with Japan. The other reason, according to Davis and Shirato [62], could be that the Chinese government would view Japan's complaint to the WTO as a hostile act and take retaliatory measures, which could do harm to Japan's business; therefore, despite China's large share of Japan's export for fishery products, Korea became the only target of Japan's complaint. Consequently, Japan could succeed in pushing other WTO Members to withdraw their import limitations on Japan's products; in fact, after Korea's loss in the WTO, China reconsidered lifting its import ban and China Hong Kong lifted its seven-year-old ban on Japanese food products in July 2018 [63].

### 3.4.2. Korea's Imposition of Precautionary Measures

In the situation where the available scientific evidence remains insufficient, the PP is advantageous to politicians who aims to combat potential risks to public health; nevertheless, it can also be a heavy burden on the politicians when there are international political-economic relations [64]. The situation of the Korean government is of the latter case, since it attempted to lift the import limitations to Japan's fishery products by announcing that "it was necessary to lift the import ban to Japan's fishery products in order to welcome the 50th Anniversary of Normalization of Korea–Japan Relations" on

15 January 2015 [65]. This statement indicated that the Korean government preferred to allow for the import of Japan's fishery products to resume as a negotiation asset to improve its diplomatic relations with Japan. In addition, as revealed in the Panel Report, because the Korean government suspended the research activities of its Civilian Expert Group, it did not fulfill its review obligation. Some Korean experts, such as Song Kee-ho, the Chairman of the International Trade Committee, suspected that the Park Geun-hye administration intended to give up the review of its provisional SPS measures in exchange for Japan's political support for its participation in the Trans-Pacific Strategic Economic Partnership (TPP) [66].

However, the passive coping attitude of the Korean government led to strong dissatisfaction from consumers; for instance, Hwang and Lee [67] revealed that 92.5% of consumers felt that they could not trust the inspection procedures conducted by the Korean government. Moreover, Kang [68] showed that 81% of respondents decreased their consumption of fishery products and would continue to avoid fishery products from Japan. These consumer concerns over the dangers stemming from Japanese fishery products have constituted political pressure on the government to some extent—there are a total of 72 national petitions addressed to Cheong Wa Dae urging the government to inspect Japanese fishery products or continue to impose import bans [69]. In order to eliminate such consumer dissatisfaction, non-governmental organizations (NGOs), the media, and members of congress held demonstrations and undertook strong lobbying activities.

First, NGOs, especially those involved in environmental issues, have long been supporters of the PP since they adopt it as a means to increase the public and stakeholders' involvement in the government's policy-making process [27]. On 21 January 2015, just one week after the public speech by the Korea's Ministry of Foreign Affairs, 23 environmental NGOs in Korea fiercely protested the government's plan to resume fishery import from Japan and lobbied to ban all food products imported from Japan [70]. On 22 May 2015, the day after Japan complained about Korea in the WTO, 10 NGOs in Korea blamed Japan's litigation and lobbied the Korean government to maintain a tough stance in order to protect its citizens from polluted fishery products. On 23 February 2018, the day after the circulation of the Panel Report, 12 NGOs protested the WTO's rulings, calling them unwarranted, and blamed the Park Geun-hye administration's ineffective response to Japan's litigation. After Korea's loss in the WTO, various NGOs organized a "Japanese Food Response Network" and increased their lobbying of the Moon Jae-in administration to maintain the precautionary SPS measures. These lobbying activities, as summarized in Figure 4, revealed that environmental NGOs have given voice to support the PP and pushed the fishery dispute into the political agenda.

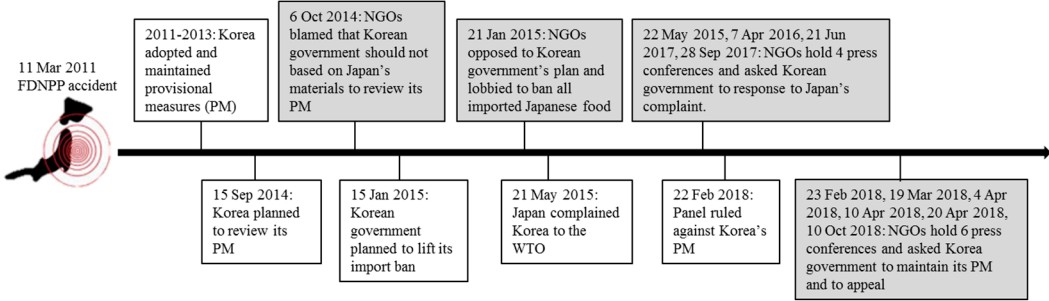

**Figure 4.** Chronology of lobbying activities of Korean NGOs. Source: Search conducted utilizing the keywords "NGO Japan Fishery products Radionuclides" in Google, last updated on 1 December 2018.

Second, most of the Korean media, being one-sided against polluted Japanese fishery products, also criticize the lack of transparency of the real radionuclide monitoring data and the incompetence of the Park Geun-hye administration. Hwang and Lee [67] revealed that the horror stories circulating in the media relating to the irreversible harm of radionuclides increased consumer worry and drove the public to demand strict regulation of Japan's fishery products. According to the calculation of Hwang and Lee [67], there were merely 50 to 100 press releases every month before the FDNPP accident, and

this number jumped to 8,323 in March 2011 immediately following the accident. The numerous press releases increased consumers' distrust of the food safety information provided by the government and have led them to impose additional pressure on Korean regulators.

Third, members of congress began to lobby more directly by pushing the Korean government to recourse to precautionary action through the National Assembly Inspection, one of the authorities of the National Assembly which shall conduct an annual inspection of overall state affairs. As shown in Figure 5, since the FDNPP accident in 2011, Korean congressmen have continued to ask the government to limit the import of fishery products from Japan.

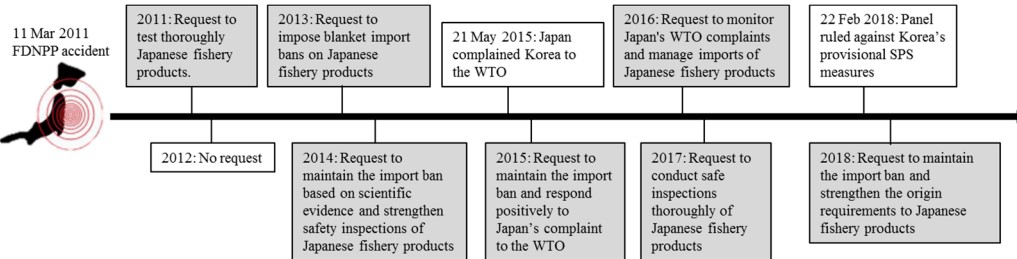

**Figure 5.** Chronology of lobbying activities of Korean congressmen. Source: Compiled by authors based on Korea's National Assembly Inspection Reports from 2011 to 2018 [71].

The persistent lobbying activities of NGOs, the media, and congressmen have led the Park Geun-hye administration to abandon its plan to lift the import ban, and pushed the Moon Jae-in administration to maintain the import limitations and appeal to the WTO.

## 4. Conclusions and Policy Recommendations

The *Korea-Radionuclides* verifies that, although the WTO ostensibly acknowledges SD as one of its far-reaching objectives, its narrow examination of Article 5.7 of the SPS Agreement remains adrift of upholding the goal of the preamble of the WTO Agreement. To date, panels and AB have taken different approaches toward the interpretation of Article 5.7 in which PP can find a reflection. Panels have not been kind to countries taking provisional SPS measures, as countries who have attempted to apply Article 5.7 in justifying their provisional measures have never succeeded in their claims. One reason for this could be the lack of an explicit endorsement of the PP in the SPS Agreement. The other reason could be the effect of the precedents—although future disputes are not bound by precedents, previous Panels/AB decisions are still given significant persuasive authority [72]. The rigorous examination of four requirements by past panels/AB has made it very difficult, if not impossible, for countries to adopt and maintain provisional measures using the language in Article 5.7. Accordingly, Members have confronted great uncertainty regarding whether their provisional measures would pass muster by the Panel and AB, and the Panel of *Korea-Radionuclides* ruled against Korea, underscoring the fact that there is still relatively little room to reconcile the PP with the SPS Agreement.

The Panel's interpretation of the PP led to Korea's failure to fulfill R1 and R2, and may cause systemic problems as follows. From a legal perspective, the continued reluctance of panels to give Members considerable policy space on the interpretation of Article 5.7 [42] leaves us in the dark in terms of the situations under which governments might successfully adopt precautionary measures. Furthermore, if the WTO Members' domestic law adopted the PP while the adjudicating bodies find their provisional measures in contravention of the SPS Agreement, governments would encounter difficulties in ensuring the quality (i.e., the offending measure is withdrawn) and timeliness (i.e., the implementing action is taken within the reasonable time period) of compliance actions [73], which would in turn undermine the authority of the WTO. From a political-economic perspective, politicians may submit to lobbying pressure and prefer to take precautionary measures to gain voter support. If the WTO rules against the respondent, political lobbying by interest groups tends to pressure the politicians to reject the implementation.

The failure of the Korean government to review its provisional measures within a reasonable period of time has led to its failure to fulfill R4. Our political-economic observations revealed that the Park Geun-hye administration intended to passively deal with Japan's complaint for international political-economic relations, and this passive attitude appeared to contribute to Korea's disadvantage in the WTO's judgment, leaving a heavy burden on the Moon Jae-in administration. Since Korea failed to comply with such review obligations, it seems that there is little possibility to reverse the decision in the appeal process.

In light of these considerations, it is advisable for the WTO Members to avoid adopting aggressive versions of the PP, because any excessive precaution may be used as a tool to please specific lobbying groups or could even be disguised as protectionism; on the other hand, a rigid interpretation style of the WTO to PP should also be avoided, since it has failed to catch up with the growing public concerns related to decision-making in the face of insufficient scientific evidence. The AB has long taken an approach of "prudent precaution" which can balance Members' rights to the protection of human health and their obligation to research and review their provisional measures [10,74,75]. This "prudent precaution" approach appears to be feasible in actual judicial decisions based on the fact that the European Court of Justice has put this approach into practice and effectively protected the environment and public health from potential risks through a number of judgments, such as the *Pfizer/Alpharma* judgments and *Vitamins* line of cases [76].

Specifically, this balance can be realized by efforts from three dimensions—the WTO adjudicating parties, the government, and the lobbying groups of importing countries. First, the WTO adjudicating parties can shift the burden of proof to the complaining party by requiring it to prove that the responding party does not fulfill R1 and R2. Moreover, it is desirable for panels to follow AB's approach of "prudence and precaution" where risks are irreversible by allowing WTO Members the policy space [77] necessary to operationalize the PP in future cases. Second, the government of the importing country should seek to design and apply constructive reviewing strategies; in doing so, it can attempt to apply new SPS measures based on the available scientific evidence, even if it loses in the panel procedure. Third, the lobbying groups of the importing country should make use of their political mobilization to push their government to fulfill R3 and R4 in order to maintain the legitimacy of their precautionary measures.

Commercial liberalization can act as one of the means to promote SD, which is acknowledged as a far-reaching goal in the WTO. The rationale for this is that commercial liberalization leads to an increase of wealth, which creates resources for better social policies and environmental management [21]. Accordingly, it is legitimate to pursue a balance between commercial liberalization and sustainable development. This study verifies the WTO's perceived indifference to SD by conducting a case analysis on *Korea-Radionuclides* from a legal approach. However, the pairing of legality and legitimacy is relevant and necessary to improve the value of the research result because the operation of the law depends on the match between them [78]. It might be desirable for future research to frame the analysis of the WTO's provisions in the logic of the quadruple helix—international organizations, scientific environment, corporate landscape, and civil societ—to ensure the legitimacy of the WTO.

**Author Contributions:** Y.C. prepared the original draft and wrote the paper; E.K. suggested research ideas and contributed to the revision of the paper.

**Funding:** This research received no external funding.

**Acknowledgments:** We are grateful to *Sustainability* editors and the anonymous reviewers of the journal for useful comments on an earlier draft of this article.

**Conflicts of Interest:** The authors declare no conflict of interest.

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
