# Peer review of "Sustainable Development in World Trade Law: Application of the Precautionary Principle in Korea-Radionuclides"

_sustainability, doi:10.3390/su11071942_

Round 1

Reviewer 1 Report

It is advisable a clearer delimitation between legality and legitimacy at WTO level. Very useful would be the framing of the analysis in the binomial provisions of the treaties - Dispute Settlement Body`s jurisprudence, the placement of the analysis in the logic of the quadruple helix - international organizations-scientific environment-corporate landscape-civil society. It is necessary to clarify the nature of the basic concepts - commercial liberalization and sustainable development. The author has to explain his / her position on these concepts by arguing which of these are goals and what are means. 

Author Response

Response to Reviewer 1 Comments

We are grateful to you for your insightful comments which have helped us improve the quality of our article.

Point 1: It is advisable a clearer delimitation between legality and legitimacy at WTO level.  Very useful would be the framing of the analysis in the binomial provisions of the treaties - Dispute Settlement Body`s jurisprudence, the placement of the analysis in the logic of the quadruple helix - international organizations-scientific environment-corporate landscape-civil society.

Response 1: Your insightful suggestions provide potential avenues of research in an interdisciplinary approach to ensure the legitimacy of the WTO, and we are interested in conducting an in-depth study on this issue. However, to re-frame the analysis “in the binomial provisions of the treaties - Dispute Settlement Body’s jurisprudence, the placement of the analysis in the logic of the quadruple helix” takes considerable time as well as interdisciplinary cooperation. Thus, we have reflected your suggestions as follows:

(Line 603-613) The pairing of legality and legitimacy is relevant because the operation of the law depends on the match between them [1]. Due to the decentralized structure of international law, the legitimacy perceived is a major factor for the WTO to operate effectively [2]. Vice versa, Members would be reluctant to comply with their obligations if they consider that binding rules of the WTO lack legitimacy [1]. The Panel’s narrow interpretation of the application of the PP in Article 5.7 of the SPS Agreement disregards the effect of WTO rules in promoting SD in a long term. It could further erode legitimacy of the WTO. There are inevitably winners and losers so legitimacy cannot depend upon a single stakeholder. Therefore, an examination of legitimacy of the WTO should be placed in the logic of the quadruple helix - international organizations, scientific environment, corporate landscape, and civil society. This provides potential avenues of future research in an interdisciplinary approach to ensure legitimacy of the WTO.

Point 2: It is necessary to clarify the nature of the basic concepts - commercial liberalization and sustainable development. The author has to explain his / her position on these concepts by arguing which of these are goals and what are means.

Response 2: According to the reviewer’s suggestion, we have added the following contents:

(Line 598-603) Commercial liberalization can act as one of means to promote SD which is acknowledged as a far-reaching goal in the WTO. The rationale for this is that commercial liberalization leads to increase of wealth which creates resources for better social policies and environmental management [3]. Accordingly, it is legitimate to pursue a balance between commercial liberalization and sustainable development. However, WTO’s perceived indifference to SD undermines its legitimacy [3].

References:

1.   Cottier, T. The Legitimacy of WTO Law; Swiss National Centre of Competence in Research: 2009; pp 1-32.

2.   Elsig, M. The World Trade Organization's Legitimacy Crisis: What Does the Beast Look Like. J. World Trade 2007, 41, 75-98.

3.   Lydgate, E.B. Sustainable Development in the WTO: From Mutual Supportiveness to Balancing. World Trade Rev. 2012, 11, 621-639.

Reviewer 2 Report

The article provides a relatively concise review of the Korea – Radionuclides panel decision of 2018. The following are meant to be constructive comments and while they are extensive, all of them are meant to be improving the article.

The first set of comments are more specific in nature, while the latter set of comments are more general.

Line 47: it remains unclear throughout the article whether SD and PP are commensurate with one another in the authors' view. This should be clarified early on.

Line 52: the interpretation of the PP is not uniform between the panels and the appellate body and the authors imply as much here and throughout the article. Other researcher has convincingly shown the different approaches between the panel level and the appellate body in this regard, see Wagner, Law Talk v. Science Talk: The Languages of Law and Science in WTO Proceedings, 35 Fordham International Law Journal 151 (2011) which reviews all the existing cases until 2011 and distinguishes the differing approaches between the panel and the appellate body levels.

Paragraph beginning in line 56: it is unclear what the connection between the different treaties are in this instance. I assume the authors want to draw the line between the UN Conference on Environment and Development and the WTO agreement. If so that should be clearer. Even if one were to follow the author's contention that ST was set up as a central principle for new negotiations in the Doha round, that does not necessarily have an impact on the legal nature of the SPS agreement. Moreover, it appears clear that the Doha round is at best comatose and more realistically dead. 

Line 66: some, should this be per placed by any? What is the threshold here is any threat or harm to the public or the environment sufficient, should include public health?

Line 67: strike precautionary

line 68 replace minimizing with limiting

line 71: replace Imission with Immission

line 75: replace in certain risks with uncertainty

line 84: concluding from analyzing three countries to a widespread adoption of the PP is problematic at best. There are plenty of studies that show a widespread adoption which the authors should cite to.

Line 94 and emerging principle of international environmental law (not laws) is not necessarily existing principle, moreover the author should make clear what the exact legal nature of these treaties mean for general international law purposes. This could consist in soft law for example. The print size question that the author should answer here, in legal terms, is what the quality of the precautionary principle is.

Line 97 according to the authors the precautionary principle has only found reflection in the SPS agreement. Others have laid down that it also finds reflection in the GATT,

Line 111: it appears that the authors had previously established that the precautionary principle is reflected in article 5.7 of the SPS agreement, but curiously they now seem to backpedal on this particular conclusion which is central to the article.

Line 118: most cases wouldn't as that is not the wording of Article 5.7 SPS Agreement

Line 150: PP and 5.7 are equated again 

Line 158: FDNPP - first time using that acronym?

Line 235: precedent is not unchangeable in the WTO, especially if the prior decision is from a panel

Line 248: replace one with party 

Line 262 and following paragraph: this is where the context of Articles 2 and 3 SPS Agreement matter - they should be explained briefly as the average reader won't be familiar with the legal context in which Article 5.7 operates. The least one should explain is Article 3.3 SPS Agreement here. 

Line 272: R1 as the authors call it can be satisfied - it is not and here I am emphatic - impossible to do. It has to be difficult in the logic of the SPS Agreement as it is a deviation from an international standard. But the AB has made it clear that WTO Members have a right to avail themselves and it is done so numerous times. 

Line 279: it can be long-running as long as new evidence is being sought; there is nothing in the case to suggest otherwise 

Line 279-280: of course it has to do that - if it is a new measure, it has to meet the requirements of the SPS Agreement, there is nothing - absolutely nothing - wrong with that! Authors may have a misunderstanding of how the SPS Agreement works? 

Line 315: judicial justice - don't understand the term 

Line 373 and following paragraph: authors would do well to be mindful of the nature of WTO remedies - in this instance the Korean measure may have achieved the desired outcome already. Remedies are prospective and are designed to bring a WTO member back into compliance. They are never retrospective, despite arguments to the contrary or the desirability of such a change in the remedies system. The time that it took to litigate appears to have been enough for Korean fisheries to substitute the fish coming from Japan so that lobbying by domestic constituencies may have been enough 

Line 433 and subsequent paragraph: authors never mention if the media was one-sided and biased against Japanese products. It appears to have been the case which is exactly what WTO law is designed against. It would be good for authors to provide context. 

Line 455: why shouldn't there be a rigorous application of the law and its standards? Maybe what the authors mean is narrow? 

457: PP and 5.7 are being equated again? 

458-459: AB has said over and over again that PP finds reflection in 5.7 SPS Agreement; also note that AB and panels have had very, very different approaches towards the SPS Agreement in general and 5.7 in particular. 

467 et seq.: the statement in second sentence of that paragraph is not true. 5.7 is not a paragon of clarity, but it is not paradoxical. The burden of proof issue is not as complicated as the authors make it out to be. 

472-473: welcome to international law, this is a statement that should be struck; there will always be allegations on sovereignty with decisions of international dispute settlement bodies, especially if they take place in a heated domestic environment as has clearly been the case in this instance 

474: the compliance rate of WTO members with DS decisions is very high - sentence needs to be revised. 

487: authors finally acknowledge the problem inherent in 5.7 - it is a fine balancing act between protectionism on the one hand and legitimate concerns on the other. 

490: AB has taken this road for quite some time, it always of course depends on cases being brought, but Wagner has shown convincingly that the AB has taken a path for governments to take if they genuinely want to address these concerns. the same is true for Article XX GATT defenses, by the way. 

503: this is a stunning statement - if governments contravene the requirement laid down in 5.7 they should of course be found in violation. AB has made clear numerous times that the four requirements are part of a package and need to fulfilled in a cumulative fashion! 

General comments: 

The article should be revised in line with the above. Additionally, the authors may want to consider consulting authors such as Epps, Gruszczynski, Howse, Foster and many others who have written on this issue for quite some time. The legal literature consulted in the article misses many of the more important authors that have provided useful insights into the jurisprudential developments in this area. 

The article touches on, but then misses out, on some of the important points that this case raises. The SPS Agreement as a whole, Article 5.7 in particular, but really all of WTO law needs to balance between competing interests and authors would do well to be more cognizant of that tension. 

Author Response

Response to Reviewer 2 Comments

We are grateful to you for your insightful comments which have helped us improve the quality of our article.

Point 1: Line 47: it remains unclear throughout the article whether SD and PP are commensurate with one another in the authors' view. This should be clarified early on.

Response 1: In our view, SD and PP are not commensurate with one another because there are a lot of means to achieve SD. PP is one of them, i.e., SD is a goal and PP is one of the means to achieve this goal. We mentioned the relationship between them in Line 29-31 as “in order to achieve SD, the PP, which is regarded as a ‘prerequisite’ or ‘key foundation’ for SD, is one of the most appropriate means that can be used [1-3].” As suggested by the reviewer, we have clarified their relationship again as follows:

(Line 48-51) Sustainable development is defined as “development which meets the needs of the present without compromising the ability of future generations to meet their own needs” [4]. There are numerous principles to ensure SD, such as principles of “user pays”, “polluter pays”, and “subsidiary principle”. The PP is one of them [5]. It can be “a valuable aid” to SD [6].  

Point 2: Line 52: the interpretation of the PP is not uniform between the panels and the appellate body and the authors imply as much here and throughout the article. Wagner has convincingly shown the different approaches between the panel level and the appellate body in this regard, see Wagner, Law Talk v. Science Talk: The Languages of Law and Science in WTO Proceedings, 35 Fordham International Law Journal 151 (2011) which reviews all the existing cases until 2011 and distinguishes the differing approaches between the panel and the appellate body levels.

Response 2:  First, we revised the sentence as follows:

(Line 55-56) Chapter 4 identifies systemic problems of the disparity between panels and AB in interpreting PP,

Second, after reading the recommended article as well as panel/AB Reports of related disputes, we recognized that we missed the differing approaches between the panel and AB. Therefore, we have added the following contents into Section 3.1:

(Line 191-208) It is worth noting that the interpretation of PP is not uniform between panels and AB. This disparity has been shown in the adjudications of SPS disputes [7]. Early in EC- Hormones, the AB had made it clear that “responsible, representative governments commonly act from perspectives of prudence and precaution where risks of irreversible, e.g., life-terminating, damage to human health are concerned” [8] (para. 124). This statement indicates that the AB appears to allow WTO Members the policy space necessary to operationalize the PP [9]. This approach of interpretation has become more evident in US/Canada Continued Suspension [7]. According to the Panel’s interpretation, the condition to “make relevant, previously sufficient, evidence now insufficient” was that “there must be a critical mass of new evidence and/or information that calls into question the fundamental precepts of previous knowledge and evidence” [10] (para. 7.648). However, the AB reversed the Panel’s finding and criticized that such requirement was too inflexible since it led to a “paradigm shift” which occurred not frequent [11] (para. 703). Furthermore, the AB pointed out that Members should be allowed to “take a provisional measure where new evidence from a qualified and respected source puts into question the relationship between the preexisting body of scientific evidence and the conclusions regarding the risks” [11] (para. 703).

Although the AB deserves credit for moving further away from a strict interpretation of Article 5.7 [9], it did not make a final finding as to the consistency or inconsistency of the EC’s provisional measures with Article 5.7 [11] (para. 736), leaving this fundamental question open [12].

Point 3: Paragraph beginning in line 56: it is unclear what the connection between the different treaties are in this instance. I assume the authors want to draw the line between the UN Conference on Environment and Development and the WTO agreement. If so that should be clearer. Even if one were to follow the author's contention that ST was set up as a central principle for new negotiations in the Doha round, that does not necessarily have an impact on the legal nature of the SPS agreement. Moreover, it appears clear that the Doha round is at best comatose and more realistically dead.

Response 3: We intended to illustrate the connection between SD and multilateral trade system by referring to Agenda 21 which was adopted in the UN Conference on Environment and Development as a background. According to reviewer’s instructions, we have clarified the connection between them as well as the legal nature of SD as follows:

(Line 60-84) International trade is a powerful ally of SD [13]. Back in 1992, Agenda 21, adopted in the United Nations Conference on Environment and Development recognized that a multilateral trading system could contribute to SD [14]. At that time, the multilateral trading system came under the General Agreement on Tariffs and Trade (GATT), the WTO’s predecessor. When the WTO was established in 1995, WTO Members included direct references to the objective of SD [13]. The preamble of the Marrakesh Agreement Establishing the World Trade Organization (the WTO Agreement) mandated that it should be “…allowing for the optimal use of the world's resources in accordance with the objective of sustainable development, seeking both to protect and preserve the environment and to enhance the means for doing so…” [15]. This mandate was reaffirmed in the 2001 Doha Ministerial Declaration which declared that trade liberalization should contribute to SD. Practically, WTO Members set up SD as a central principle for new negotiations throughout all Doha negotiations [16]. Although the stated importance indicates that SD wields great influence, due to the non-binding nature of preambular language and the comatose state of the Doha Round, SD does not form a legal rule. It is only a general principle of the WTO’s legal framework [17]. This is not to say SD has no formal influence [17]. Instead, it not only serves as a guidance for the implementation of WTO agreements, such as the SPS Agreement, but also serves as a justification for decisions by dispute settlement bodies [18].

Point 4: Line 66: some, should this be per placed by any? What is the threshold here is any threat or harm to the public or the environment sufficient, should include public health?

Response 4: First, we have replaced “some” with “any”, see Line 86. Second, according to Wagner [9], the PP is not limited to the realm of the environment. It also includes public health. see Line 87.

Point 5: Line 67: strike precautionary

Response 5: We removed “precautionary”, see Line 87.

Point 6: line 68 replace minimizing with limiting

Response 6: We replaced “minimizing” with “limiting”, see Line89.

Point 7: line 71: replace Imission with Immission

Response 7: We replaced “Imission” with “Immission”, see Line 92.

Point 8: line 75: replace in certain risks with uncertainty

Response 8: We replaced “uncertain risks” with “uncertainty”, see Line 97.

Point 9: line 84: concluding from analyzing three countries to a widespread adoption of the PP is problematic at best. There are plenty of studies that show a widespread adoption which the authors should cite to.

Response 9: We cited several prior studies [19-21] to show a “widespread adoption” of the PP in Line 107. Moreover, we listed more countries as examples:

(Line 104-106) More examples of the adoption of PP can be found in the domestic laws of Australia, Canada, New Zealand, Chile, India, Pakistan, and so on [6].

Point 10: Line 94 and emerging principle of international environmental law (not laws) is not necessarily existing principle, moreover the author should make clear what the exact legal nature of these treaties mean for general international law purposes. This could consist in soft law for example. The print size question that the author should answer here, in legal terms, is what the quality of the precautionary principle is.

Response 10: First, we made clear the legal nature of the international instruments in Line 107-119. Second, since there is no consensus regarding the legal status of the PP as a general principle of law (we mentioned this early on in the Introduction section, see Line 33-39), we decided to delete the controversial words “emerging principle of international environmental law”.

(Line 107-119) The widespread adoption [19-21] of the PP in these domestic and regional laws has contributed to its incorporation into numerous international instruments of both soft and hard law [22]. The most prominent statement of the PP in a soft law instrument [9] can be found in Principle 15 of the Rio Declaration on Environment and Development of 1992: “In order to protect the environment, the precautionary approach shall be widely applied by States according to their capabilities. Where there are threats of serious or irreversible damage, lack of full scientific certainty shall not be used as a reason for postponing cost-effective measures to prevent environmental degradation” [23]. Since then, the application of PP has proliferated a vast number of hard law instruments such as the United Nations Framework Convention on Climate Change, Convention for the Protection of the Marine Environment of the North-East Atlantic, the Cartagena Protocol on Biosafety to the Convention on Biological Diversity, and the Stockholm Convention on Persistent Organic Pollutants [22].

Point 11: Line 97 according to the authors the precautionary principle has only found reflection in the SPS agreement. Others have laid down that it also finds reflection in the GATT, see for example Wagner, Taking Interdependence Seriously: Reassessing the Precautionary Principle in International Trade Law, 20 Cardozo Journal of International and Comparative Law 713 (2012).

Response 11: We referred to the recommended article of Wagner (2012) and revised this paragraph as follows:

(Line 129-148) The SPS Agreement provides for an entry point for PP in Article 5.7, albeit with qualifications [9]. Article 5.7 of the SPS Agreement allows for the adoption (in the first sentence) and maintenance (in the second sentence) of provisional SPS measures provided that the following four requirements are fulfilled: (i) a Member may adopt provisional measures in such cases where “relevant scientific evidence is insufficient” to conduct a risk assessment (R1); (ii) the provisional measures must be adopted on the basis of “available pertinent information” (R2); (iii) the Member maintaining the provisional measures shall “seek to obtain the additional information necessary for a more objective assessment of risk” (R3); and (iv) the Member maintaining the provisional measures shall review that measure “within a reasonable period of time” (R4). This provisional approach to avoid risks with insufficient scientific evidence is similar to the “better safe than sorry” wisdom of the PP, which underpins uncertainty and irreversibility [24].

Not only the SPS Agreement, but also Article XX of GATT, appear to take account of elements of the PP [9]. For example, in EC–Asbestos, the AB opened an entry point for elements of the PP under Article XX (b) of GATT and allowed a high degree of deference to domestic decision-making on appropriate level of risk. It found that WTO Members had the right to determine the level of protection of health that they considered appropriate [25]. However, a further analysis on the application of PP in the GATT is not within the scope of this study. Thus, the following sections will only focus on the application of PP in Article 5.7 of the SPS Agreement.

Point 12: Line 111: it appears that the authors had previously established that the precautionary principle is reflected in article 5.7 of the SPS agreement, but curiously they now seem to backpedal on this particular conclusion which is central to the article.

Response 12: We removed the contradictory part and revised the statement as follows:

(Line 151-153) The WTO adjudicating bodies have examined the PP on a case-by-case basis [22] and WTO jurisprudence on it is as follows.

Point 13: Line 118: most cases wouldn't as that is not the wording of Article 5.7 SPS Agreement

Response 13: We revised the sentence as suggested by the reviewer:

(Line 159-160) most cases did not mention PP as it is not in the wording of Article 5.7 of the SPS Agreement.

Point 14: Line 150: PP and 5.7 are equated again

Response 14: We revised the statement as follows:

(Line 209-211) Here, we conduct a case analysis on Korea-Radionuclides for the following reasons: first, it is the latest case related to Article 5.7, so a study on it can contribute to a useful understanding of the current state of WTO ruling on Article 5.7

Point 15: Line 158: FDNPP - first time using that acronym?

Response 15: We shortened the “Fukushima Dai-ichi Nuclear Power Plant” to “FDNPP”, see Line 216.

Point 16: Line 235: precedent is not unchangeable in the WTO, especially if the prior decision is from a panel, again see Wagner, Law Talk

Response 16: We read the recommended articles and found that we made a mistake here. Therefore, we have removed the wrong statement (see Line 296-299). We thank the reviewer for pointing this out.

Point 17: Line 248: replace one with party

Response 17: We replaced “one” with “party”, see Line 309-310.

Point 18: Line 262 and following paragraph: this is where the context of Articles 2 and 3 SPS Agreement matter - they should be explained briefly as the average reader won't be familiar with the legal context in which Article 5.7 operates. The least one should explain is Article 3.3 SPS Agreement here.

Response 18: According to reviewer’s instructions, we have explained the legal context in which Article 5.7 operates as follows:

(Line 323-335) Before analyzing rulings of the Panel, it is worth explaining the legal context in which Article 5.7 operates. First, Article 3.3 of the SPS Agreement confers upon Members the right to determine their own level of protection, including “a higher level of sanitary or phytosanitary protection than…international standards, guidelines or recommendations”, under the condition that “there is a scientific justification, or as a consequence of the level of sanitary or phytosanitary protection a Member determines to be appropriate in accordance with the relevant provisions of paragraphs 1 through 8 of Article 5”. Second, while Article 2.1 of the SPS Agreement allows Members to take SPS measure, Article 2.2 mandates that this can be done with a considerable amount of scrutiny [7] by requiring Members to ensure that SPS measure “is applied only to the extent necessary…and is not maintained without sufficient scientific evidence, except as provided for in paragraph 7 of Article 5”. Thus, Article 5.7 which allows Members to adopt provisional SPS measures in cases where “relevant scientific evidence is insufficient" operates as a “qualified exemption” from the obligation under Article 2.2 [26] (para. 80).

Point 19: Line 272: R1 as the authors call it can be satisfied - it is not and here I am emphatic - impossible to do. It has to be difficult in the logic of the SPS Agreement as it is a deviation from an international standard. But the AB has made it clear that WTO Members have a right to avail themselves and it is done so numerous times.

Response 19: We revised the statement as follows:

(Line 345-350) It is very difficult for Korea to satisfy R1 because in the logic of the SPS Agreement, it is a deviation from an international standard. Although the AB of precedents has repeated its long-standing position that Members have right to set appropriate level of protection[11] (para. 692), the Panel’s ruling against Korea in this case highlights that the justification for the adoption of precautionary measures is still difficult to exercise in reality.

Point 20: Line 279: it can be long-running as long as new evidence is being sought; there is nothing in the case to suggest otherwise

Response 20: We revised the sentence as follows:

(Line 355-357) However, Korea’s provisional measures can be long-running as long as new evidence is being sought.

Point 21: Line 279-280: of course it has to do that - if it is a new measure, it has to meet the requirements of the SPS Agreement, there is nothing - absolutely nothing - wrong with that! Authors may have a misunderstanding of how the SPS Agreement works?

Response 21: We removed this inappropriate statement, see Line 357-358. We thank the reviewer for pointing this out.

Point 22: Line 315: judicial justice - don't understand the term

Response 22: We removed this term and revised the sentence as follows:

(Line 392-394) The following three reasons may contribute to Japan’s complaint: possible economic benefits, domestic lobbying pressure, and the political warning effect to other countries.

Point 23: Line 373 and following paragraph: authors would do well to be mindful of the nature of WTO remedies - in this instance the Korean measure may have achieved the desired outcome already. Remedies are prospective and are designed to bring a WTO member back into compliance. They are never retrospective, despite arguments to the contrary or the desirability of such a change in the remedies system. The time that it took to litigate appears to have been enough for Korean fisheries to substitute the fish coming from Japan so that lobbying by domestic constituencies may have been enough

Response 23: We agree with the reviewer that the WTO remedies are prospective rather than retrospective. However, in this paragraph (Line 451-466), we have intended to explain the third reason, i.e., the political warning effect on other countries, of Japan’s WTO complaint only against Korea. Our opinion is shared by Allee [27] who has conducted an empirical study on the dispute initiation in WTO.

Point 24: Line 433 and subsequent paragraph: authors never mention if the media was one-sided and biased against Japanese products. It appears to have been the case which is exactly what WTO law is designed against. It would be good for authors to provide context

Response 24: We have added the following statement according to the Reviewer’s suggestion:

(Line 511-515) most of Korean media, being one-sided against polluted Japanese fishery products, also criticize the lack of transparency of the real radionuclide monitoring data and the incompetence of the Park Geun-hye administration.

Point 25: Line 455: why shouldn't there be a rigorous application of the law and its standards? Maybe what the authors mean is narrow?

Response 25: We replaced “rigorous” with “narrow”, see Line 535.

Point 26: 457: PP and 5.7 are being equated again?

Response 26: We revised the sentence as follows:

(Line 539-540) as countries who have attempted to apply Article 5.7 in justifying their provisional measures have never succeeded in their claims.

Point 27: 458-459: AB has said over and over again that PP finds reflection in 5.7 SPS Agreement; also note that AB and panels have had very, very different approaches towards the SPS Agreement in general and 5.7 in particular.

Response 27: We thank the reviewer for pointing this out. We have modified this sentence as follows:

(Line 536-540) To date, panels and AB have taken different approaches toward the interpretation of Article 5.7 in which PP can find a reflection. Panels have not been kind to countries taking provisional SPS measures as countries who have attempted to apply Article 5.7 in justifying their provisional measures have never succeeded in their claims.

Point 28: 467 et seq.: the statement in second sentence of that paragraph is not true. 5.7 is not a paragon of clarity, but it is not paradoxical. The burden of proof issue is not as complicated as the authors make it out to be.

Response 28: We removed the second sentence and revised the statement as follows:

(Line 550-553)  From a legal perspective, the continued reluctance of panels to give Members considerable policy space on the interpretation of Article 5.7 [7] leaves us in the dark in terms of the situations under which governments might successfully adopt precautionary measures.

Point 29: 472-473: welcome to international law, this is a statement that should be struck; there will always be allegations on sovereignty with decisions of international dispute settlement bodies, especially if they take place in a heated domestic environment as has clearly been the case in this instance

Response 29: According to reviewer’s instructions, we have removed the statement: “supporters of the PP may perceive the judgment as an infringement on their sovereignty”, see Line 555-556. We thank the reviewer for pointing this out.

Point 30: 474: the compliance rate of WTO members with DS decisions is very high - sentence needs to be revised.

Response 30: As suggested by the reviewer, we have revised the sentence as follows:

(Line 556-559) governments would encounter difficulties in ensuring the quality (i.e. the offending measure is withdrawn) and timeliness (i.e. the implementing action is taken within the reasonable time period) of compliance actions [28]  

Point 31: 487: authors finally acknowledge the problem inherent in 5.7 - it is a fine balancing act between protectionism on the one hand and legitimate concerns on the other.

Response 31: Yes, this is what we intended to express, see Line 570-574.

Point 32: 490: AB has taken this road for quite some time, it always of course depends on cases being brought, but Wagner has shown convincingly that the AB has taken a path for governments to take if they genuinely want to address these concerns. the same is true for Article XX GATT defenses, by the way.

Response 32: First, we revised this sentence as follows. Second, AB’s approach on this issue was added in Section 3.1, Line 191-205. Third, the application of PP in Article XX GATT was added in Section 2, Line 142-148.  

(Line 575-576) The AB of precedents has long taken an approach of “prudent precaution”

(Line 191-205) It is worth noting that the interpretation of PP is not uniform between panels and AB. This disparity has been shown in the adjudications of SPS disputes [7]. Early in EC- Hormones, the AB had made it clear that “responsible, representative governments commonly act from perspectives of prudence and precaution where risks of irreversible, e.g., life-terminating, damage to human health are concerned” [8] (para. 124). This statement indicates that the AB appears to allow WTO Members the policy space necessary to operationalize the PP [9]. This approach of interpretation has become more evident in US/Canada Continued Suspension [7]. According to the Panel’s interpretation, the condition to “make relevant, previously sufficient, evidence now insufficient” was that “there must be a critical mass of new evidence and/or information that calls into question the fundamental precepts of previous knowledge and evidence” [10] (para. 7.648). However, the AB reversed the Panel’s finding and criticized that such requirement was too inflexible since it led to a “paradigm shift” which occurred not frequent [11] (para. 703). Furthermore, the AB pointed out that Members should be allowed to “take a provisional measure where new evidence from a qualified and respected source puts into question the relationship between the preexisting body of scientific evidence and the conclusions regarding the risks” [11] (para. 703).

 (Line 142-148) Not only the SPS Agreement, but also Article XX of GATT, appear to take account of elements of the PP [9]. For example, in EC–Asbestos, the AB opened an entry point for elements of the PP under Article XX (b) of GATT and allowed a high degree of deference to domestic decision-making on appropriate level of risk. It found that WTO Members had the right to determine the level of protection of health that they considered appropriate [25]. However, a further analysis on the application of PP in the GATT is not within the scope of this study. Thus, the following sections will only focus on the application of PP in Article 5.7 of the SPS Agreement.

Point 33: 503: this is a stunning statement - if governments contravene the requirement laid down in 5.7 they should of course be found in violation. AB has made clear numerous times that the four requirements are part of a package and need to fulfilled in a cumulative fashion!

Response 33: We have made corrections according to the Reviewer’s comments:

(Line 590-593) it is desirable for panels to follow AB’s approach of “prudence and precaution” where risks are irreversible by allowing WTO Members the policy space [29] necessary to operationalize the PP in future cases.

References:

1.         Norton, B. Sustainability, Human Welfare, and Ecosystem Health. Environ. Value 1992, 1, 97-111.

2.         Paterson, J. Sustainable Development, Sustainable Decisions and the Precautionary Principle. Nat. Hazards 2007, 42, 515-528.

3.         Som, C.; Hilty, L.M.; Köhler, A.R. The Precautionary Principle as a Framework for a Sustainable Information Society. J. Bus Ethics 2009, 85, 493-505.

4.         Report of the World Commission on Environment and Developmene: Our Common Future, United Nations, 1987. Availabe online: https://sswm.info/sites/default/files/reference_attachments/UN%20WCED%201987%20Brundtland%20Report.pdf (accessed on 20 March 2019).

5.         Harris, J.M. Basic Principles of Sustainable Development; Global Development and Environment Institute: United States, 2000; pp 21-41.

6.         Trouwborst, A. Evolution and Status of the Precautionary Principle in International Law, 1st ed.; Kluwer Law International: New York, US, 2002.

7.         Wagner, M. Law Talk v. Science Talk: The Languages of Law and Science in WTO Proceedings. Fordham Int'l L. J. 2011, 35, 151-200.

8.         WTO Appellate Body Report. European Communities—Measures Concerning Meat and Meat Products (Hormones) [EC-Hormones]; WT/DS26/AB/R and WT/DS48/AB/R, adopted 16 January 1998.

9.         Wagner, M. Taking Interdependence Seriously: The Need for a Reassessment of the Precautionary Principle in International Trade Law. Cardozo J. Int'l & Comp. L. 2011, 20, 713-769.

10.       WTO Panel Report. United States—Continued Suspension of Obligations in the EC—Hormones Dispute [US—Continued Suspension]; WT/DS320/R, adopted 31 March 2008.

11.       WTO Appellate Body Report. United States—Continued Suspension of Obligations in the EC—Hormones Dispute [US—Continued Suspension]; WT/DS320/AB/R, adopted 16 October 2008.

12.       Vecchione, E. Is it Possible to Provide Evidence of Insufficient Evidence? The Precautionary Principle at the WTO. Chi. J. Int'l L. 2012, 13, 153-178.

13.       Sustainable Development in the WTO. Availabe online: https://www.wto.org/english/tratop_e/envir_e/sust_dev_e.htm (accessed on 13 March 2019).

14.       United Nations Conference on Environment & Development Agenda 21, United Nations, Availabe online: https://sustainabledevelopment.un.org/content/documents/Agenda21.pdf (accessed on 16 January 2019).

15.       Agreement Establishing the World Trade Organization, World Trade Organization, 1995. Availabe online: https://www.wto.org/english/docs_e/legal_e/04-wto.pdf (accessed on 21 January 2019).

16.       Gehring, M.W.; Segger, M.C.C. Sustainable Development in World Trade Law; Kluwer Law International: Hague, Netherlands, 2005; pp. 27-70.

17.       Lydgate, E.B. Sustainable Development in the WTO: From Mutual Supportiveness to Balancing. World Trade Rev. 2012, 11, 621-639.

18.       Aseeva, A. (Un) Sustainable Development (s) in International Economic Law: A Quest for Sustainability. Sustainability 2018, 10, 4022.

19.       Zander, J. The Application of the Precautionary Principle in Practice: Comparative Dimensions, 1st ed.; Cambridge University Press: New York, US, 2010.

20.       Gardiner, S.M. A Core Precautionary Principle. J. Polit. Philos. 2006, 14, 33-60.

21.       Trouwborst, A. Prevention, Precaution, Logic and Law: The Relationship between the Precautionary Principle and the Preventative Principle in International Law and Associated Questions. Erasmus L. Rev. 2009, 2, 105-127.

22.       Sirinskiene, A. The Status of Precautionary Principle: Moving towards a Rule of Customary Law. Jurisprudence 2009, 4, 349-364.

23.       Rio Declaration on Environment and Development, United Nations, 1992. Availabe online: http://www.un.org/documents/ga/conf151/aconf15126-1annex1.htm (accessed on 21 January 2019).

24.       Jiangyuan, F.; Blennerhassett, J. Is Article 5.7 of the SPS Agreement an Application of the Precautionary Principle? Frontiers L. China 2015, 10, 268-294.

25.       WTO Appellate Body Report. European Communities—Measures Affecting Asbestos and Products Containing Asbestos [EC—Asbestos]; WT/DS135/AB/R, adopted 12 March 2001.

26.       WTO Appellate Body Report. Japan—Measures Affecting Agricultural Products [Japan-Agricultural Products II]; WT/DS76/AB/R, adopted 22 February 1999.

27.       Allee, T. Developing Countries and the Initiation of GATT/WTO Disputes. In Proceedings of annual meeting of the International Studies Association. Honolulu, HI.

28.       Davey, W.J. Compliance Problems in WTO Dispute Setlement. Cornell Int'l L.J. 2009, 42, 119-128.

29.       Epps, T. Reconciling Public Opinion and WTO Rules under the SPS Agreement. World Trade Rev. 2008, 7, 359-392.

Round 2

Reviewer 1 Report

Added points and clarifications have led to an increase in the quality of the work, in this form meeting the requirements for publication.

Author Response

Response to Reviewer 1 Comments

Point 1: Added points and clarifications have led to an increase in the quality of the work, in this form meeting the requirements for publication.

Response 1: We are grateful to you for your insightful comments which have helped us improve the quality of our article.

Reviewer 2 Report

line 78: is any threat really captured or does PP only kick in when a particular threshold is met? 

line 329: reword "Although the AB of precedents has" ... maybe: "Although the AB has"?

line 331: author seems to hold the view that R1 cannot be satisfied, but the facts of the case appear - at least this reviewer - to give credence to the panel's findings. It is true that under the logic of the SPS Agreement, 5.7 is difficult to implement exactly for the reason that otherwise governments would have an easy way to abuse the right to rely on the PP 

line 550: "The AB of precedents has long taken an approach" - please fix 

574/575: "Due to the decentralized structure of international law, the legitimacy

575 perceived is a major factor for the WTO to operate effectively [79]." odd sentence and comes out of the blue. This is an entirely different debate in international law that does not appear necessary to touch here. 

577/578/579: "The Panel’s narrow interpretation of the application of the PP in Article 5.7 of the 578 SPS Agreement disregards the effect of WTO rules in promoting SD in a long term. It could further 579 erode legitimacy of the WTO." repeats statement two or three sentences prior. 

Note: the entire last paragraph as well as some of the other insertions give the reader the feeling that they were very quickly written. They should be revised in terms of substance and grammar/ style. The author only very lightly touched on the question of legitimacy prior to this paragraph and opens a new front without it being necessary. I suggest revising the argument in this paragraph or - preferentially - deleting the paragraph in its entirety. It does not flow well nor does it add much -  because of its superficial nature - to the thrust of the article. 

please ensure that grammar and style is corrected. The inserted versions contain a number of errors, such a s missing articles (the / a) and punctuation mistakes. 

Author Response

(We also provided our response as a PDF file.)

Response to Reviewer 2 Comments

We are grateful to you for your insightful comments which have helped us improve the quality of our article.

Point 1: line 78: is any threat really captured or does PP only kick in when a particular threshold is met?  

Response 1: We replaced “any” with “serious or irreversible” (See Line 86-87: “generally, the PP is defined as the concept in which, when an activity causes serious or irreversible threat of harm [1-4] to the environment or public health [2], measures can be taken even in situations of scientific uncertainty [5].”) for the following reasons:

(1) There is neither a singular universally accepted definition of the PP nor a consensus on the threshold for triggering it [4,6]. To clarify the threshold of the PP, we attempt to compare various definitions of PP in relevant international instruments in Table 1.

Table 1. Threshold of PP in International Instruments

Source

Definition & Threshold

UNGA   Resolution the World Charter for Nature (1982)

“Activities   which are likely to cause irreversible   damage to nature shall be avoided”; “Activities which are likely to pose   a significant risk to nature shall   be preceded by an exhaustive examination; their proponents shall demonstrate   that expected benefits outweigh potential   damage to nature, and where potential adverse effects are not fully   understood, the activities should not proceed”

London   Declaration: Second International Conference on the Protection of the North   Sea (1987)

“…in   order to protect the North Sea from possibly   damaging effects of the most dangerous substances, a precautionary   approach is necessary…”

Houston   Economic Summit Declaration, G-7 Meeting (1990)

“In   the face of threats of irreversible   environmental damage, lack of full scientific certainty is no excuse to   postpone actions which are justified in their own right.”

Bergen   Ministerial Declaration on Sustainable Development (1990)

“Where   there are threats of serious or   irreversible damage, lack of full scientific certainty should not be used   as a reason for postponing measures to prevent environmental degradation.”

Rio   Declaration on Environment and Development (1992)

“Where   there are threats of serious or   irreversible damage, lack of full scientific certainty shall not be used   as a reason for postponing cost-effective measures to prevent environmental   degradation.”

UN   Framework Convention on Climate Change (1992)

“Where   there are threats of serious or   irreversible damage, lack of full scientific certainty should not be used   as a reason for postponing such measures”

Convention   on Biological Diversity (1992)

“...Noting also that where there is a threat of significant reduction or loss of biological diversity,   lack of full scientific certainty should not be used as a reason for   postponing measures to avoid or minimize such a threat…”

Wingspread   Statement on the Precautionary Principle (1998)

“…it   is necessary to implement the Precautionary Principle: When an activity   raises threats of harm to human   health or the environment, precautionary measures should be taken even if   some cause and effect relationships are not fully established   scientifically.”

EU   Communication (2000)

“The   precautionary principle applies where scientific evidence is insufficient,   inconclusive or uncertain and preliminary scientific evaluation indicates   that there are reasonable grounds for concern that the potentially dangerous effects on the environment, human, animal   or plant health may be inconsistent with the high level of protection chosen   by the EU.”

UNESCO (2005)

“When human activities may lead to morally unacceptable harm that is scientifically plausible but   uncertain, action shall be taken to avoid or diminish that harm”

Sources: Compiled by authors based on the studies of Wagner [2], Trouwborst [7], and Zander [8].

(2)  The various definitions/thresholds of the PP mentioned above are often categorized into a weak PP and a strong version of PP. According to Som, et al. [9], in the weak version of the PP, “precautionary measures are taken only where major, irreversible risks could occur and their scientific level of proof is high. In addition, only precautionary measures that have low costs may be taken.” On the other hand, in the strong version of the PP, “precautionary measures should be taken whenever there is any speculative evidence of a risk. Neither does the risk have to be high nor irreversible. Precautionary measures are taken irrespective of their costs.” Table 2 shows a comparison of the two versions of PP.

Table 2. Weak PP vs Strong PP

Weak PP

Strong PP

How   mandatory is the application of the PP?

Precautionary   measures may be taken where

Precautionary   measures must be taken where

Extent   of threat

Major/serious/irreversible   risks might exist

Minor/reversible   risks might exist

Extent   of uncertainty

The scientific level of proof is high

Only speculative evidence   exists

Extent   of action

The costs for precautionary   measures must be low

The costs   for precautionary measures may be high

Advocators

Sunstein [10], Wiener [11], etc.

Kriebel, et al. [12], Sachs [4], etc.

Source: Compiled by authors based on the studies of Som, Hilty and Köhler [9] and Sachs [4].

(3) Although there are controversial opinions regarding the threshold of the PP, Di Salvo and Raymond [3] have attempted to analyse the PP in the search for a dominant formulation among 238 articles in a variety of disciplines. Their meta-analysis of various expressions of the PP has shown that dominant use of the PP has become weaker over time and broadly resembles Principle 15 of the 1992 Rio Declaration. Therefore, we decided to replace the inappropriate word “any” with “serious or irreversible” and cited several prior studies that share the same opinion with us. We thank the reviewer for pointing this out.

Point 2: line 329: reword "Although the AB of precedents has" ... maybe: "Although the AB has"?

Response 2:  As suggested by the reviewer, we replaced “Although the AB of precedents has” with “Although the AB has”, see Line 347.

Point 3: line 331: author seems to hold the view that R1 cannot be satisfied, but the facts of the case appear - at least this reviewer - to give credence to the panel's findings. It is true that under the logic of the SPS Agreement, 5.7 is difficult to implement exactly for the reason that otherwise governments would have an easy way to abuse the right to rely on the PP

Response 3: We agree with the reviewer that it is not impossible to satisfy R1. Therefore, we removed the inappropriate statement and revised the sentence as follows. We thank the reviewer for pointing this out.

(Line 347-354) Although the AB has repeated its long-standing position that Members have right to set appropriate level of protection[13] (para. 692) and according to Wagner [1,2], the AB has allowed WTO Members to “rely on minority scientific opinions when determining whether there was insufficient scientific evidence in order to justify more stringent measures”, the Panel’s ruling highlights the continued reluctance of the Panel to give governments considerable discretion to justify their provisional measures when there is insufficient evidence to conduct a risk assessment [1].

Point 4: line 550: "The AB of precedents has long taken an approach" - please fix

Response 4: We revised this sentence as “The AB has long taken an approach”, see Line 579.

Point 5: 574/575: "Due to the decentralized structure of international law, the legitimacy 575 perceived is a major factor for the WTO to operate effectively [79]." odd sentence and comes out of the blue. This is an entirely different debate in international law that does not appear necessary to touch here.

577/578/579: "The Panel’s narrow interpretation of the application of the PP in Article 5.7 of the 578 SPS Agreement disregards the effect of WTO rules in promoting SD in a long term. It could further 579 erode legitimacy of the WTO." repeats statement two or three sentences prior.

The entire last paragraph as well as some of the other insertions give the reader the feeling that they were very quickly written. They should be revised in terms of substance and grammar/ style. The author only very lightly touched on the question of legitimacy prior to this paragraph and opens a new front without it being necessary. I suggest revising the argument in this paragraph or - preferentially - deleting the paragraph in its entirety. It does not flow well nor does it add much -  because of its superficial nature - to the thrust of the article.

Response 5: According to reviewer’s instructions, we have deleted the odd and repeated statement “Due to the decentralized structure … for the WTO to operate effectively” and “The Panel’s narrow interpretation erode legitimacy of the WTO. We thank the reviewer for pointing this out.

However, the pairing of legality and legitimacy is relevant because the operation of the law depends on the match between them [14]. Moreover, Reviewer 1 has recommended taking into account legitimacy of pursuing SD in the WTO and has suggested potential avenues of future research in an interdisciplinary approach to improve the value of research result. Therefore, instead of deleting the last paragraph in its entirety, we revised it as follows:

(Line 602-612) Commercial liberalization can act as one of means to promote SD which is acknowledged as a far-reaching goal in the WTO. The rationale for this is that commercial liberalization leads to increase of wealth which creates resources for better social policies and environmental management [15]. Accordingly, it is legitimate to pursue a balance between commercial liberalization and sustainable development. This study verifies WTO’s perceived indifference to SD by conducting a case analysis on Korea-Radionuclides from a legal approach. However, the pairing of legality and legitimacy is relevant and necessary to improve the value of the research result because the operation of the law depends on the match between them [14]. It might be desirable for future researches to frame the analysis of WTO’s provisions in the logic of the quadruple helix - international organizations, scientific environment, corporate landscape, and civil society to ensure legitimacy of the WTO.

References:

1.         Wagner, M. Law Talk v. Science Talk: The Languages of Law and Science in WTO Proceedings. Fordham Int'l L. J. 2011, 35, 151-200.

2.         Wagner, M. Taking Interdependence Seriously: The Need for a Reassessment of the Precautionary Principle in International Trade Law. Cardozo J. Int'l & Comp. L. 2011, 20, 713-769.

3.         Di Salvo, C.P.; Raymond, L. Defining the Precautionary Principle: An Empirical Analysis of Elite Discourse. Environ. Polit. 2010, 19, 86-106.

4.         Sachs, N.M. Rescuing the Strong Precautionary Principle from its Critics. U. Ill. L. Rev. 2011, 2011, 1285-1338

5.         Morris, M. The Precautionary Principle: Good for Environmental Activists, Bad for Business. J.Bus. Adm. 2010, 9, 1-24.

6.         Sandin, P. A Paradox out of Context: Harris and Holm on the Precautionary Principle. Camb. Q. Healthc. Ethic. 2006, 15, 175-183.

7.         Trouwborst, A. Evolution and Status of the Precautionary Principle in International Law, 1st ed.; Kluwer Law International: New York, US, 2002.

8.         Zander, J. The Application of the Precautionary Principle in Practice: Comparative Dimensions, 1st ed.; Cambridge University Press: New York, US, 2010.

9.         Som, C.; Hilty, L.M.; Köhler, A.R. The Precautionary Principle as a Framework for a Sustainable Information Society. J. Bus Ethics 2009, 85, 493-505.

10.       Sunstein, C.R. Laws of fear: Beyond the Precautionary Principle, 1st ed.; Cambridge University Press: Cambridge, UK, 2005.

11.       Wiener, J.B. Precaution in a Multi-Risk World. In Human and Ecological Risk Assessment: Theory and Practice Paustenbach, D.J., Ed. Wiley Interscience: New York, US, 2002.

12.       Kriebel, D.; Tickner, J.; Epstein, P.; Lemons, J.; Levins, R.; Loechler, E.L.; Quinn, M.; Rudel, R.; Schettler, T.; Stoto, M. The Precautionary Principle in Environmental Science. Environ. Health Persp. 2001, 109, 871-876.

13.       WTO Appellate Body Report. United States—Continued Suspension of Obligations in the EC—Hormones Dispute [US—Continued Suspension]; WT/DS320/AB/R, adopted 16 October 2008.

14.       Cottier, T. The Legitimacy of WTO Law; Swiss National Centre of Competence in Research: 2009; pp 1-32.

15.       Lydgate, E.B. Sustainable Development in the WTO: From Mutual Supportiveness to Balancing. World Trade Rev. 2012, 11, 621-639.
